# Effects of Exercise on Positive Symptoms, Negative Symptoms, and Depression in Patients with Schizophrenia: A Systematic Review and Meta-Analysis

**DOI:** 10.3390/ijerph20043719

**Published:** 2023-02-20

**Authors:** Myoungsuk Kim, Yongmi Lee, Hyunju Kang

**Affiliations:** College of Nursing, Kangwon National University, Chuncheon-si 24341, Republic of Korea

**Keywords:** schizophrenia, exercise, positive symptoms, negative symptoms, depression, meta-analysis

## Abstract

This study was performed to evaluate the effects of exercise on positive and negative symptoms and depression in patients with schizophrenia through a systematic review and meta-analysis focusing on randomized controlled trials (RCTs). PubMed, Embase, CINAHL, MEDLINE, Cochrane Library, PsycINFO, and Web of Science were searched from their inception to 31 October 2022. We also conducted a manual search using Google Scholar. This meta-analysis was conducted according to the PRISMA guidelines. The methodological quality of the studies was assessed using the Cochrane risk-of-bias tool for randomized trials. To identify the cause of heterogeneity, subgroup analysis, meta-ANOVA, and meta-regression analyses were performed as moderator analyses. Fifteen studies were included. The meta-analysis (random-effects model) for overall exercise showed a medium significant effect (standardized mean difference [SMD] = −0.51, 95% confidence interval [CI]: −0.72 to −0.31) on negative symptoms, a small significant effect (SMD = −0.24, 95% CI: −0.43 to −0.04) on positive symptoms, and a nonsignificant effect (SMD = −0.87, 95% CI: −1.84 to 0.10) on depression. Our findings demonstrate that exercise can relieve the negative and positive symptoms of schizophrenia. However, the quality of some included studies was low, limiting our results for clear recommendations.

## 1. Introduction

Approximately 1% of the global population suffers from schizophrenia. Schizophrenia is a persistent mental health problem worldwide, and long-term care and management are crucial throughout life [1,2]. Patients with schizophrenia experience various psychiatric problems, including positive symptoms, such as delusions or hallucinations, and negative symptoms, such as apathy, isolation, or decreased social functioning [3].

Positive symptoms can cause violent behavior and influence patients’ social interactions and daily functioning [4]. These factors are also associated with social stigma and a high hospitalization rate [5]. Negative symptoms can occur at any stage in schizophrenia [6]. Previous studies have shown that negative symptoms play a role in and contribute greatly to disability in patients with schizophrenia [7]. Moreover, positive and negative symptoms not only affect patients themselves, but also burden the families who take care of these patients [5].

Patients with schizophrenia frequently have comorbid mental illness, such as depression [8]. The prevalence of depressive disorders in schizophrenia is reportedly approximately 40%. Previous studies have demonstrated that the prevalence of various depressive symptoms in schizophrenia ranges from 6% to 75% [6]. Depressive symptoms can interact with the clinical symptoms of schizophrenia, especially the negative symptoms [9], with comorbid depression possibly associated with suicide in these patients [10].

Therefore, health providers have tried to treat and care for the positive and negative symptoms of schizophrenia, in addition to other psychological problems such as depression. In clinical practice, pharmacological approaches, including antipsychotics, are the first-line treatments, and they have been used to treat clinical symptoms in patients with schizophrenia. However, a study on the effectiveness of antipsychotics for treating schizophrenia reported symptom reduction of only approximately 60% following antipsychotic treatment [11]. Positive symptoms of schizophrenia are effectively treated, but the negative symptoms are more difficult to diminish with antipsychotics [12]. Moreover, these medications result in side-effects such as weight gain and metabolic syndrome [13], and many patients are reluctant to take them because of these side effects.

Although the main treatment for schizophrenia is antipsychotic medication, other modalities such as cognitive therapy and exercise have been proposed as adjuvant treatments [3]. Nonpharmacological strategies such as physical activity lead to psychiatric and physical health problems in patients with schizophrenia. Physical activity can improve cognition, and socio-occupational functioning could provide valuable adjunctive treatment for schizophrenia [14,15]. Physical exercise, such as aerobic exercise and yoga, is considered an adjuvant treatment for schizophrenia [16,17] and may be required to achieve better outcomes [18].

Several studies have shown that physical activity can significantly improve positive and negative symptoms and other psychological symptoms in this population [19]. In contrast, lower physical activity has been associated with greater negative symptoms, and reduced exercise capacity has been associated with more severe negative and depressive symptoms [20].

Evidence indicates that physical activity can affect symptoms of schizophrenia. Low levels of brain-derived neurotropic factor (BDNF) are related to schizophrenia symptoms. There may be a remarkable association between psychotic symptoms and BDNF levels in the cognitive functioning areas [21]. In addition, the positive symptoms of schizophrenia have been associated with excessive dopamine activity in the mesolimbic pathway, while the negative symptoms have been associated with reduced dopamine activity in the prefrontal cortex; dopamine dysregulation may induce symptoms of schizophrenia [3]. Physical activities may improve the symptoms of schizophrenia by increasing neurotrophic and neuroprotective mechanisms [18,22], and increased BDNF can mediate the effects of exercise through synaptic plasticity [3,21].

The hippocampus is a brain region associated with depression, and physical activity may contribute to a decrease in depressive symptoms by inducing an increase in BDNF levels in the hippocampus [23]. Furthermore, according to the biopsychosocial model [24], biologic, psychologic, and social processes function interdependently and reciprocally, and they are all integral to and are interactively involved in health and illness. According to this model, the mechanisms interacting at the organism, interpersonal, and environmental levels result in illness. Therefore, as physical activities are based on biological, social, or psychological factors, they may affect symptoms related to schizophrenia [21].

Previous systematic reviews and meta-analyses have investigated the effects of physical activity on the positive and negative symptoms in patients with schizophrenia. However, some of these reviews included both randomized controlled trials (RCTs) and non-RCT studies [22], and some focused on specific exercises [25] or only on negative symptoms [26]. Furthermore, these previous reviews also included patients with other mental disorders or with psychosis [27]. Therefore, the present study attempted to include only recently published RCTs and focus on the positive and negative symptoms of schizophrenia and depressive symptoms, which is an additional psychological issue in individuals with schizophrenia.

This study aimed to determine the effect of physical activity on positive symptoms and negative symptoms, alongside depression, one of the most common mental health issues in people with schizophrenia, through a systematic review and meta-analysis of controlled trials. We also explored the impact of various intervention characteristics that affect the outcomes of physical activity. The findings of this study can be used as basic data for developing interventions to manage symptoms and psychological problems in people with schizophrenia.

## 2. Methods

### 2.1. Eligibility Criteria

The eligibility criteria for this review were as follows: (1) participants: patients aged 18 years or older who met the diagnostic criteria for schizophrenia or schizoaffective disorder; (2) interventions: any type of exercise (aerobic, resistance, multicomponent, and neuromotor exercise); (3) intervention in the control group: usual care, no intervention, and stretching; (4) outcomes: positive symptoms, negative symptoms, and depression; (5) study design: RCTs only. We excluded studies that combined other types of interventions with exercise, studies that evaluated exercise without a control group, studies that included participants with schizophrenia and other diagnoses, and studies with missing data essential for the meta-analysis. We included all studies published in English between 2011 and 31 October 2022. This review was performed in accordance with the Preferred Reporting Items for Systematic Reviews and Meta-Analyses (PRISMA) guidelines. The review protocol was registered in the International Prospective Register of Systematic Reviews (PROSPERO) (CRD42022318841).

### 2.2. Search Strategies

We performed a literature search in the following international databases in the period of 18 September 2022 to 31 October 2022: PubMed, Embase, Cumulative Index of Nursing and Allied Health Literature (CINAHL), MEDLINE, Cochrane Central Register of Controlled Trials, PsycINFO, and Web of Science. Additionally, to identify literature that was not identified through the search strategy, we conducted a manual search on Google Scholar.

A search strategy was formulated using Medical Subject Headings (MeSH) and Embase Subject Headings database (Emtree) terms, synonyms, free text, and truncations. The search terms were the following: (schizophrenia or schizoaffective disorder), (exercise), and (positive symptoms or negative symptoms, or depression), as well as derivatives of these terms (Appendix A).

### 2.3. Study Selection and Data Extraction

Two reviewers independently searched the databases and performed the eligibility assessments. The titles and abstracts of the studies were screened according to the eligibility criteria. The full text of the studies was reviewed, and eligible studies were selected. We then extracted the data independently using a standard data extraction form, which included information on the authors, publication year, country, study design, sample size, and intervention (type of exercise, outcomes, duration, sessions per week, time, comparator, and follow-up). Discrepancies during inclusion of studies in the final review were resolved by discussion between the two reviewers or, if unresolved, by a third reviewer.

### 2.4. Risk-of-Bias Assessment and Quality Appraisal

This study assessed the risk of bias using the revised Cochrane risk-of-bias tool (Cochrane Training, London, UK) for randomized trials (ROB 2) [28]. Two reviewers independently performed assessments and determined the ROB rating. Any discrepancies in ratings were resolved through discussion among the authors.

### 2.5. Data Synthesis

The meta-analyses were performed using the R software package Meta version 4.0.3 (R Foundation for Statistical Computing, Vienna, Austria). We used a random-effects model assuming that the true effect samples may differ across studies. The effect size was estimated on the basis of the standardized mean difference (SMD; Hedges’ ĝ) with a 95% confidence interval (CI). The effect size was represented according to Cohen’s classification (small < 0.40, medium = 0.40 to 0.70, large > 0.70) [29]. We analyzed the heterogeneity by χ^2^ and *I*^2^ statistics. An *I*^2^ value greater than 50% and a *p*-value less than 0.1 indicated substantial heterogeneity according to the criteria of the Cochrane Handbook for Systematic Reviews of Intervention. In addition, the interpretation of the *I*^2^ value depends on many factors, but a rough guide for the same in a meta-analysis of randomized trials is as follows: 0–40%, might not be important; 30–60%, may represent moderate heterogeneity; 50–90%, may represent substantial heterogeneity; 75–100%, considerable heterogeneity [28]. To investigate the cause of heterogeneity, we performed subgroup analysis according to the exercise type, duration, and session. Funnel plots and Egger’s regression tests were used to identify publication bias. The pooled effect size after excluding the study with the greatest contribution to the pooled effect size was evaluated through sensitivity analysis.

## 3. Results

### 3.1. Study Selection

A total of 1109 studies were identified in the database and manual search. After removing 816 duplicate articles, the titles and abstracts of 286 studies were screened, and 246 studies were excluded. Next, 40 studies were sought for retrieval, and six studies were excluded as they were not retrieved. After excluding 23 studies that did not meet the inclusion criteria, 11 studies were reviewed. Meanwhile, we identified seven additional studies from websites and through a manual search of citation; after excluding three of them, four were included in the review. Finally, we selected 15 studies in the systematic review (Figure 1) [1,17,18,19,30,31,32,33,34,35,36,37,38,39,40].

### 3.2. Characteristics of the Studies

Fifteen RCTs were included in this study. The characteristics of the included studies are presented in Table 1. Most of these studies were performed in Asia (eight studies): Japan (four studies), Taiwan (two studies), India (one study), and Hong Kong (one study). The remaining studies were performed in the United States of America (two studies), Serbia (one study), Greece (one study), Iran (one study), the Netherlands (one study), and Brazil (one study). A total of 1642 participants were randomized to the exercise groups (*n* = 884) and control groups (*n* = 758). On analyzing the negative and positive symptoms, the withdrawal rate was found to be 16.67% in the exercise group and 11.68% in the control group; according to the depression analysis, it was 11.87% in the exercise group and 12.65% in the control group.

In the intervention group, the types of exercises included aerobic exercise in 10 studies, neuromotor exercise in four studies, and multicomponent exercise in three studies according to the criteria presented by the American College of Sports Medicine (ACSM) [41]. Exercise duration was 20 weeks or more in three studies, 12 weeks in seven studies, and 8 weeks in five studies. The sessions per week of exercise included 1–2 sessions per week in eight studies, 3–4 sessions per week in five studies, and five sessions or more per week in two studies. Participants in the control group received standard pharmacological therapy, routine care, standard treatment, stretching, or no intervention. The effectiveness of exercise in the improvement of negative or positive symptoms was assessed in 15 studies, whereas that on the reduction of depression was assessed in four studies (Table 1).

The outcomes of negative and positive symptoms in 15 studies were measured using the Positive and Negative Syndrome Scale (PANSS). The outcome of depression was measured using the Chinese version of the PANSS, Montgomery and Åsberg Depression Scale (MADRS), and Calgary Depression Scale for Schizophrenia (CDSS).

### 3.3. Risk of Bias in Included Studies

The findings of the risk-of-bias (RoB) assessment are presented in Figure 2A. Eleven RCTs were judged to have a low RoB, and four studies had some concerns overall. Concerns regarding the randomization process were identified in three studies, which had insufficient information regarding the random allocation sequence. Concerns related to outcome measurement were identified in one study as it was unclear whether the outcome assessors were aware of the interventions. Overall, the RoB assessment of the 15 included studies showed that 73.3% were at low risk, 26.7% had some concerns, and none were at high risk (Figure 2B).

### 3.4. Results of Individual Studies and Synthesis of Results

We conducted a basic meta-analysis for each outcome: negative symptoms, positive symptoms, and depression.

#### 3.4.1. Negative Symptoms

As shown in Table 1, the studies of Behere et al. [30] and Ho et al. [1] can be presented by dividing each study into two experimental groups and two control groups. Therefore, although 15 studies were included, more than 15 studies were analyzed. Seventeen RCTs reported the effects of exercise on the negative symptoms (Figure 3). The exercise group showed better improvement in negative symptoms than the control group in seven studies. The largest effect size was −1.35. The SMD (Hedges’ ĝ) was −0.51 (95% CI: −0.72 to −0.31), thereby showing that exercise can improve negative symptoms among schizophrenia patients.

According to the criteria of the Cochrane Handbook for Systematic Reviews of Intervention [28], 45% of the included studies presented moderate heterogeneity (*I*^2^ = 45.0%, *p* = 0.02). Therefore, we conducted a subgroup analysis and meta-analysis of variance to investigate the potential causes of heterogeneity. First, a subgroup analysis based on the effect of negative symptoms by exercise type revealed a mean effect (Hedges’ ĝ) of −0.68 (95% CI: −1.19 to −0.16) for multicomponent exercise, −0.53 (95% CI: −0.82 to −0.23) for aerobic exercise, and −0.41 (95% CI: −0.84 to 0.03) for neuromotor exercise. However, the type of neuromotor exercise did not significantly improve the negative symptoms in patients with schizophrenia. The meta-analysis of variance showed no significant differences across the types of exercises (Q = 0.62, df = 2, *p* = 0.733).

Furthermore, we performed subgroup analyses based on exercise duration. In the subgroup analysis based on duration of exercise, the mean effects (Hedges’ ĝ) were −0.60 (95% CI: −0.88 to −0.31) for exercise duration of 8 weeks, −0.55 (95% CI: −0.98 to −0.13) for 20 weeks or more, and −0.33 (95% CI: −0.52 to −0.15) for 12 weeks. The meta-analysis of variance showed no significant differences in the duration of exercise (Q = 0.71, df = 2, *p* = 0.701). In addition, meta-regression indicated that the duration of exercise did not account for the heterogeneity of the mean effect (0%). Lastly, we performed a subgroup analysis based on exercise session. In the subgroup analysis based on sessions of exercise, the mean effects (Hedges’ ĝ) were −1.03 (95% CI: −1.39 to −0.68) for 13–24 times, −0.46 (95% CI: −1.39 to −0.68) for 25–48 times, −0.44 (95% CI: −0.75 to −0.13) for 49 times or more, and −0.10 (95% CI: −1.35 to 0.14) for 12 times or less. Twelve or fewer sessions did not significantly improve negative symptoms. In addition, a meta-analysis of variance showed significant differences between the exercise sessions (Q = 17.67, df = 3, *p* < 0.001). Moreover, the meta-regression indicated that the exercise session significantly explained 100% of the variation in the mean effects (QM = 17.67, df = 3, *p* < 0.001).

#### 3.4.2. Positive Symptoms

Seventeen RCTs reported on the effects of exercise on positive symptoms (Figure 4). The exercise group showed better improvement in positive symptoms than the control group in four studies. The largest effect size was −1.87. The SMD (Hedges’ ĝ) was −0.24 (95% CI: −0.43 to −0.04), thereby showing that exercise can improve positive symptoms among schizophrenia patients. The heterogeneity between the studies was moderate (*I^2^* = 41.0%, *p* = 0.04). First, to investigate the cause of heterogeneity, we conducted a subgroup analysis by exercise type, which showed that the mean effect (Hedges’ ĝ) was −0.61 (95% CI: −1.08 to −0.14) for multicomponent exercise. Only the types of multicomponent exercise significantly improved positive symptoms among patients with schizophrenia. In addition, the meta-analysis of variance showed no significant differences between the types of exercise (Q = 3.00, df = 2, *p* = 0.222).

In addition, we performed subgroup analyses based on exercise duration. In the subgroup analysis based on duration of exercise, the mean effects (Hedges’ ĝ) were −0.79 (95% CI: −1.28 to −0.30) for exercise duration of 20 weeks or more, −0.15 (95% CI: −0.39 to 0.08) for 12 weeks, and −0.11 (95% CI: −0.43 to 0.20) for 8 weeks. Only an exercise duration of 20 weeks or more significantly improved positive symptoms among patients with schizophrenia. However, the meta-analysis of variance showed no significant differences in the duration of exercise (Q = 5.92, df = 2, *p* = 0.051).

#### 3.4.3. Depression

Four RCTs reported on the effects of exercise on depression (Figure 5). The exercise group showed a better reduction in depression than the control group in two studies. The greatest effect size was −2.12 14. The SMD (Hedges’ ĝ) was −0.87 (95% CI: −1.84 to 0.10), thereby indicating that exercise cannot reduce depression among patients with schizophrenia.

#### 3.4.4. Publication Bias

The funnel plot prepared to identify potential publication bias for negative symptoms was asymmetrical, and its lower right area appeared empty (Figure 6A). Therefore, Egger’s regression test was performed to accurately test the relationship between effect size and standard error. The *p*-value of the regression model (t = −3.49, df =15, *p* = 0.003) indicated statistical significance and, thus, a potential publication bias. To confirm the effect of this publication bias on our study outcomes, we used the trim-and-fill method. After correcting the asymmetrical funnel plot to be symmetric, including six studies that were assumed to be omitted (Figure 6B), the mean effect size reduced from −0.51 (95% CI: −0.72 to −0.30) to −0.28 (95% CI: −0.52 to −0.04). This implies that the effect size of negative symptoms may have been overestimated by −0.23 due to publication bias. However, although the effect size reduced, exercise still showed a significant effect on the improvement of negative symptoms in patients with schizophrenia.

For positive symptoms, the funnel plot of the standard error and effect sizes seemed symmetrical (Figure 6C). However, we performed an Egger’s regression test for accuracy. The *p*-value of the regression model (t = −1.89, df =15, *p* = 0.077) was not statistically significant, indicating a significant relationship between the effect size and standard error. The results confirmed a statistically significant improvement in positive symptoms among patients with schizophrenia. Regarding depression, we did not examine for publication bias because the power of the test was low due to the small number of studies.

#### 3.4.5. Sensitivity Analysis

A sensitivity analysis was performed to investigate whether within-study bias influenced the mean effect of exercise on negative symptoms (Figure 7A). After excluding the study by Khonsari [42], the overall effect size dropped to −0.46, but there was no significant difference. Therefore, the results on the effectiveness of exercise in enhancing negative symptoms in schizophrenia are considered reliable. Moreover, the sensitivity analysis results regarding whether the within-study biases affected the mean effect of exercise on negative symptoms showed that, after excluding the study by Andrade and Silva et al. [31], the overall effect size dropped to −0.17, showing a slight difference. However, in other studies, the mean effect of exercise on positive symptoms was similar, at −0.24, confirming the consistency of the overall effect size (Figure 7B).

## 4. Discussion

Schizophrenia is a chronic illness often associated with an incomplete recovery of clinical symptoms. Some symptoms of schizophrenia are difficult to resolve with antipsychotics alone [12]. Clinical symptoms of schizophrenia are related to neuronal plasticity and cognitive mechanism disorders in the brain [21,42]. Physical exercise induces an increase in new neurons in the hippocampus [22]. In patients with schizophrenia, physical exercise can increase the hippocampus size and serum BDNF levels, which can further improve the symptoms of schizophrenia [3].

This study was a systematic review and meta-analysis conducted to confirm the effect of exercise intervention on positive symptoms, negative symptoms, and depression in the schizophrenia population. Our review included 15 studies designed only as RCTs, and all 15 studies were included in the meta-analysis. Although previous meta-analyses were conducted to evaluate the effect of exercise in patients with schizophrenia, some included both controlled and uncontrolled studies [22] or selected specific exercises such as mind-body exercise, aerobic exercise, and resistance training [25]. Another study focused only on the negative symptoms of schizophrenia [26]. Furthermore, previous review studies did not include recently published RCT studies until 2022 [43]. By contrast, this study attempted to include all recently published RCTs until 2022 without being limited to specific exercises and attempted to analyze the effects of such exercise intervention not only on negative symptoms and positive symptoms, but also on depression. As such, our study differs from previous ones in those mentioned aspects.

Negative symptoms in schizophrenia are defined as the absence or diminution of normal behaviors and functions and are more difficult to treat than positive symptoms using antipsychotic medications [12,44]. Some patients are concerned about the side-effects of the medications, such as weight gain [13]. The present results support the use of nonpharmacological adjunctive exercise treatment to improve the negative symptoms of schizophrenia. In our analysis, a medium, yet significant, effect of exercise intervention on negative symptoms in schizophrenia was found (SMD = −0.51, 95% CI: −0.72 to −0.31). This was slightly higher than the effect size (SMD = 0.49) of previous research, which included both controlled and uncontrolled studies [22]. Considering that the uncontrolled studies had difficulty fully confirming the effects of experimental interventions, the results of our study, targeting only RCTs, support the effect of exercise intervention on negative symptoms in schizophrenia. Although Sabe, Kaiser, and Sentissi [26] also analyzed RCTs only, they showed a small effect size. This differs from our study in that it only dealt with physical intervention, excluding mind–body exercises such as yoga, tai chi, or complex interventions such as lifestyle coaches [26]. Our study included aerobic exercise, multicomponent exercise with aerobic and resistance, and neuromotor exercises such as yoga and tai chi. Our subgroup analysis showed that the effect size of multicomponent exercise was −0.68, that of aerobic exercise was −0.53, and that of neuromotor exercise was −0.41, which were not statistically significant. In the study by Vogel et al. [25], the effect size of aerobic exercise was small (SMD = 0.34) but that of mind–body exercises like yoga or tai chi was medium (SMD = 0.46). Both our study and Vogel et al.’s study [25] had high heterogeneity of neuromotor exercises and mind–body exercises; therefore, it should be considered that the effect size may vary depending on the diversity of the interventions. Furthermore, some Chinese studies that confirmed the effect of Tai Chi exercise alone in patients with schizophrenia showed a large effect size (SMD = −0.87) [45]. This seems to be the result of including all tai chi studies that were not included in our study, which had Chinese language restrictions. Mind–body exercises, such as yoga and tai chi, improve long-term memory as well as physical, emotional, and spiritual balance [3,45]. Future studies should evaluate the underlying mechanisms and effects of mind–body exercise on the negative symptoms of schizophrenia.

Next, in this study, the effect of exercise intervention on the positive symptoms of schizophrenia was found to have a small effect size (SMD = −0.24, 95% CI: −0.43 to −0.04). The systematic review by Dauwan et al. [22], which included both controlled and uncontrolled studies, showed that the effect of exercise intervention on positive symptoms was small (SMD = 0.32). However, the effect size of exercise on positive symptoms in Dauwan et al. [22] was higher than that in our study. Therefore, our study, which included only RCTs, confirmed that the effect of exercise on positive symptoms is relatively small. This is comparable to the results of Sabe et al. [26], who studied the effect of overall exercise intervention on positive symptoms (SMD = 0.18). Because of subgroup analysis based on the type of exercise, the effect size of aerobic and neuromotor exercises was not significant (SMD = −0.61) but was found to be only in multicomponent exercise. Therefore, it was confirmed that the multicomponent exercise intervention with aerobic and resistance exercises had a major effect on positive symptoms. However, a previous study demonstrated that the effect of aerobic exercise was significant, but that of nonaerobic exercise was not significant [26]. In our results, only aerobic exercise was not significant, but multicomponent intervention with aerobic and resistance exercises was effective. Therefore, a study comparing the effect of exercise with nonaerobic, aerobic, and other interventions is required. The positive symptoms of schizophrenia have been associated with excessive dopamine activity in the mesolimbic pathway [3]. Future research should examine whether exercise improves positive symptoms through neurotransmitter modulation in patients with schizophrenia. BDNF, the levels of which increase during physical exercise, has been shown to mediate the effects of exercise through synaptic plasticity actions for axonal and dendritic remodeling [3,21,46].

Lastly, in this study, the effect of exercise on depression in schizophrenia was not significant (SMD = −0.87, 95% CI: −1.84 to 0.10), whereas other studies showed a significant effect (SMD = 0.71) [22]. Since this previous study included both RCT and non-RCT studies, it is necessary to conduct a meta-analysis of RCTs alone to confirm the effect of exercise interventions considering the depression characteristics and symptoms of schizophrenia patients. Exercises have been suggested to have an antidepressant effect on patients with major depression disorder (MDD), which, similar to schizophrenia, is associated with decreased synaptic plasticity [3]. Increased BDNF levels in the hippocampus following physical activity may contribute to a decrease in depressive symptoms [23]. Therefore, the effect of exercise on depression in schizophrenia and depressive symptoms in MDD can be compared through a meta-analysis including a large number of studies.

Our analyzed studies had at least 30 min/day of exercise time and ranged from one to seven times per week. Most exercise interventions followed the recommendations of ACSM [41]. However, future studies are needed to identify appropriate and effective exercise guidelines for patients [19]. We confirmed that an exercise duration of 20 weeks or more had a moderate effect size in relieving the positive and negative symptoms of schizophrenia. Therefore, long-term exercise intervention is recommended to relieve clinical symptoms in patients with schizophrenia.

The findings of this review indicate that exercise is effective in improving the negative and positive symptoms of schizophrenia, which is associated with difficulty in cognition and social functioning. This can be explained by the biopsychosocial model, according to which physical activity affects an individual’s biological, social, and psychological factors [21,24].

Eight of our selected studies were performed in Asia. The cultural background and attitudes of participants in the exercise intervention could not be excluded, and this should be considered when interpreting our results.

Our study had some limitations. Firstly, the percentage of heterogeneity ranged from moderate to high. Although meta-analysis and subgroup analyses were attempted, they did not account for this heterogeneity. This could be explained by the high variety of the different exercise protocols. Secondly, some of the included studies were concerning for their methodological quality of randomized controlled trials. Thirdly, not all studies were included in our analysis because of language limitations. Lastly, it is difficult to be sure whether the changes in symptoms and depression scores were the result of exercise interventions only because all confounding factors related to experimental studies were not excluded.

## 5. Conclusions

This study aimed to evaluate the effects of exercise on positive symptoms, negative symptoms, and depression in patients with schizophrenia through a systematic review and meta-analysis focusing on RCTs, including recently updated studies. Our findings suggest that exercise interventions, including aerobic, multicomponent, or neuromotor exercises, can help improve clinical negative and positive symptoms. In particular, multicomponent exercise intervention combined with aerobic and resistance exercises had a moderate effect size in improving the positive and negative symptoms of schizophrenia. In addition, an exercise duration of 20 weeks or more had a moderate effect size in relieving the positive and negative symptoms. We recommend future reviews focusing on the level of clinical symptoms in patients with schizophrenia. It is proposed to compare the effect size between physical exercises and mind-body exercises on the symptoms of schizophrenia.

## Figures and Tables

**Figure 1 ijerph-20-03719-f001:**
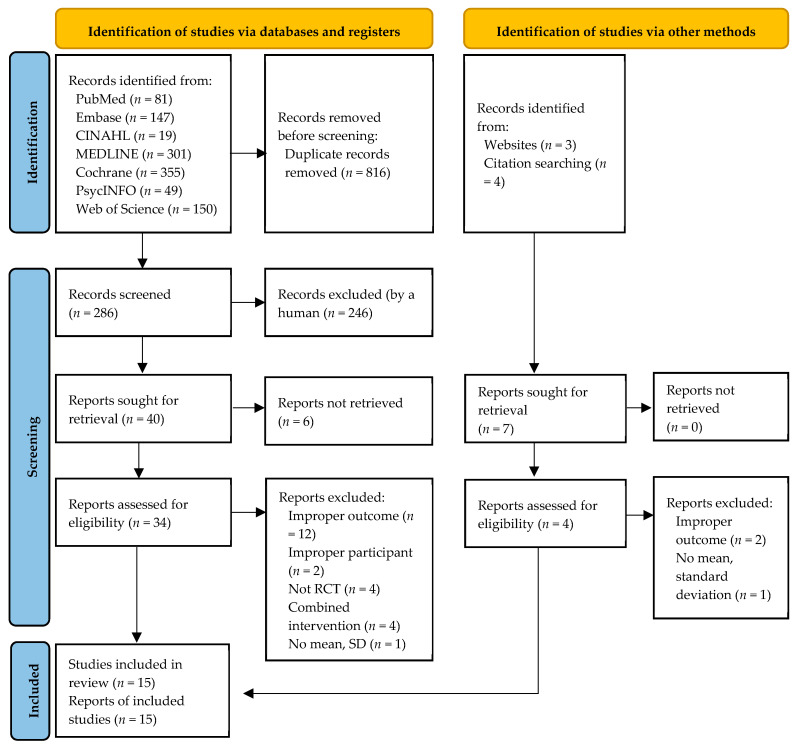
Flowchart summarizing the process of study selection.

**Figure 2 ijerph-20-03719-f002:**
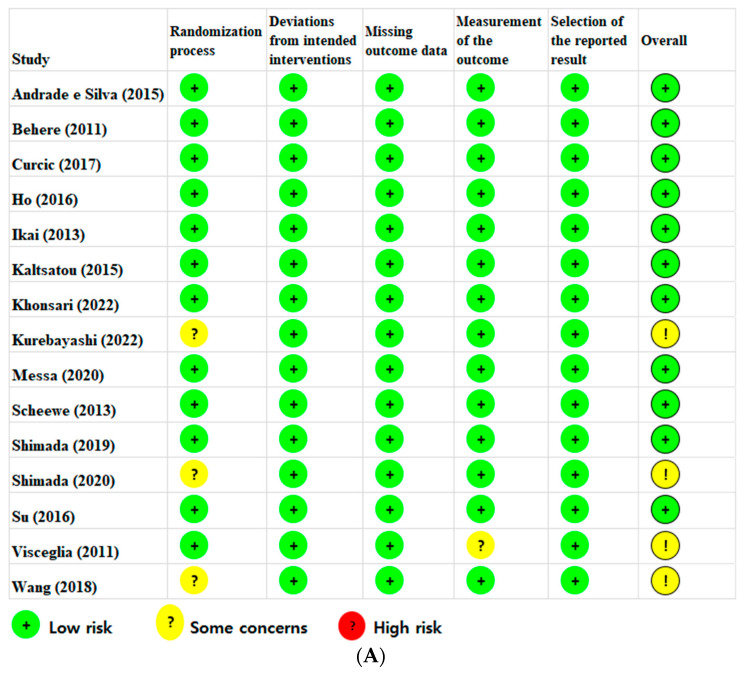
(**A**) Risk-of-bias summary according to the revised Cochrane risk-of-bias tool for randomized trials (ROB 2) [1,17,18,19,30,31,32,33,34,35,36,37,38,39,40]. (**B**) Risk-of-bias graph according to the revised Cochrane risk-of-bias tool for randomized trials (ROB 2).

**Figure 3 ijerph-20-03719-f003:**
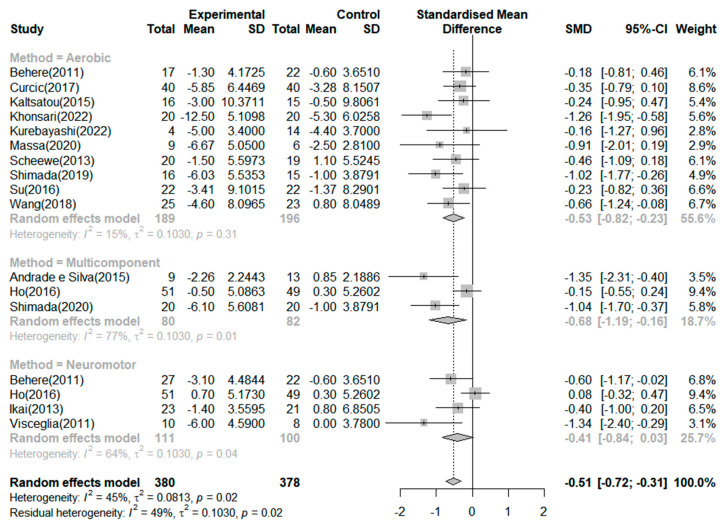
Forest plot of negative symptoms. Note: Box size represents study weighting. Diamond represents overall effect size and 95% Cis [1,17,18,19,30,31,32,33,34,35,36,37,38,39,40].

**Figure 4 ijerph-20-03719-f004:**
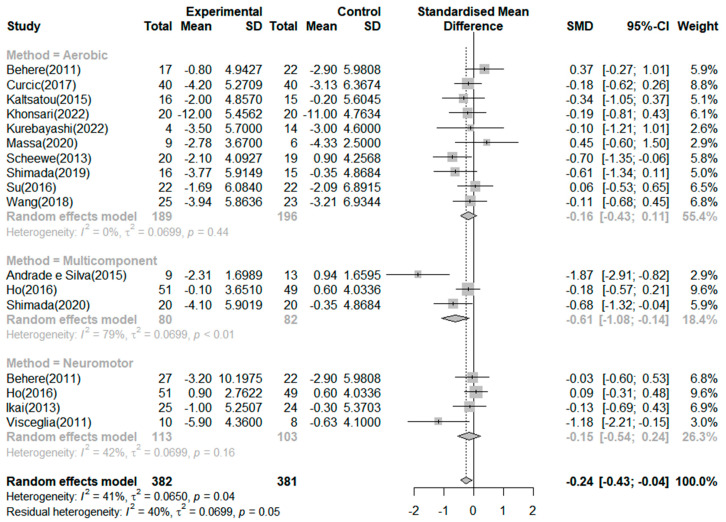
Forest plot of positive symptoms. Note: Box size represents study weighting. Diamond represents overall effect size and 95% Cis [1,17,18,19,30,31,32,33,34,35,36,37,38,39,40].

**Figure 5 ijerph-20-03719-f005:**
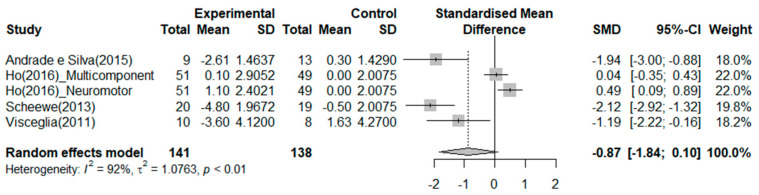
Forest plot of depression. Note: Box size represents study weighting. Diamond represents overall effect size and 95% Cis [1,17,31,38].

**Figure 6 ijerph-20-03719-f006:**
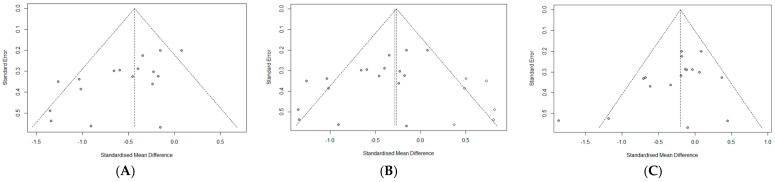
(**A**). Funnel plot for the relationship between exercise and negative symptom. (**B**)**.** Funnel plot after including other studies via the trim-and-fill method. (**C**). Funnel plot for the relationship between exercise and positive symptom.

**Figure 7 ijerph-20-03719-f007:**
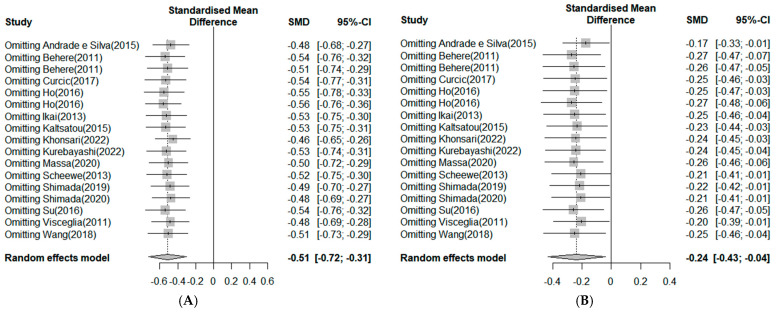
(**A**). Sensitivity analysis in negative symptoms. Note: Box size represents study weighting. Diamond represents overall effect size and 95% Cis [1,17,18,19,30,31,32,33,34,35,36,37,38,39,40]. (**B**) Sensitivity analysis in positive symptoms. Note: Box size represents study weighting. Diamond represents overall effect size and 95% Cis [1,17,18,19,30,31,32,33,34,35,36,37,38,39,40].

**Table 1 ijerph-20-03719-t001:** Characteristics of included studies on the effect of exercise.

No.	First Author (Year), Country	Study Design	Sample Size R = E:C; A = E:C	Intervention	Outcome (Instrument)	Duration (Weeks)	Sessions/Week	Time (Min)	Comparator	Follow Up
1	Andrade e Silva (2015), Brazil [31]	RCT	R = 17:16; A = 9:13	Multicomponent (concurrent exercise)	NS, PS (PANSS), depression (CDSS)	20	2	60	Minimum loading exercise	10th, 20th weeks
2	Behere (2011), India [30]	RCT	E1: R = 31:16; A = 17:22 E2: R = 34:26; A = 27:22	E1: Aerobic (brisk walking, jogging) E2: Neuromotor (yoga)	NS, PS (PANSS)	8	7	60	No intervention	2nd, 4th month
3	Curcic (2017), Serbia [32]	RCT	R = 40:40; A = 40:40	Aerobic (walking and running)	NS, PS (PANSS)	12	4	45	Standard pharmacological therapy	Post intervention
4	Ho (2016), Hong Kong [1]	RCT, non- blind	E1: R = 51:51; A = 51:49 E2: R = 51:51; A = 51:49	E1: Multicomponent E2: Neuromotor (Tai-chi)	NS, PS, depression (Chinese version of the PANSS)	12	1	60	Routine care	3rd, 6th month
5	Ikai (2013), Japan [33]	RCT, single blind	R = 25:25; A = 23:21	Neuromotor (yoga)	NS, PS (PANSS)	8	1	60	Regular day-care	Post intervention
6	Kaltsatou (2015), Greece [34]	RCT	R = 16:15; A = 16:15	Aerobic (dancing)	NS, PS (PANSS)	32	3	60	No intervention	Post intervention
7	Khonsari (2022), Iran [35]	RCT	R = 20:20; A = 20:20	Aerobic (cycling, jumps, jogging)	NS, PS (PANSS)	8	3	30	Standard pharmacological therapy	Post intervention
8	Kurebayashi (2022), Japan [36]	RCT	R = 5:17; A = 4:14	Aerobic	NS, PS (PANSS)	8	2	60	Treatment as usual	Post intervention
9	Massa (2020), USA [37]	RCT	R = 21:17; A = 9:6	Aerobic (stationary bicycle ergometer)	NS, PS (PANSS)	12	3	45	Stretching and balance training.	Post intervention
10	Scheewe (2013), Netherlands [38]	RCT	R = 31:32; A = 20:19	Aerobic (cardio-respiratory fitness)	NS, PS (PANSS), depression (MADRS)	24	2	60	Occupational therapy	Post intervention
11	Shimada (2019), Japan [19]	RCT	R = 21:17; A = 16:15	Aerobic (treadmill, stationary bike)	NS, PS (PANSS)	12	2	60	Treatment as usual	Post intervention
12	Shimada (2020), Japan [39]	RCT	R = 20:21; A = 20:20	Aerobic (treadmill, stationary bike)	NS, PS (PANSS)	12	2	60	Treatment as usual	6th, 12th month
13	Su (2016), Taiwan [40]	RCT, single blind	R = 30:27; A = 22:22	Aerobic (treadmill)	NS, PS (PANSS)	12	3	40	Stretching and toning	3rd, 6th month
14	Visceglia (2011) USA [17]	RCT	R = 10:8; A = 10:8	Neuromotor (yoga)	NS, PS, depression (PANSS)	8	2	45	No intervention	Post intervention
15	Wang (2018), Taiwan [18]	RCT, single blind	R = 33:29; A = 25:23	Aerobic	NS, PS (PANSS)	12	5	40	Stretching	Post intervention

Note. RCT, randomized controlled trial; E, experimental group; C, control group; R, randomized; A, analyzed; NS, negative symptoms; PS, positive symptoms; PANSS, Positive and Negative Syndrome Scale; MADRS, Montgomery and Åsberg Depression Scale; CDSS, Calgary Depression Scale.

## Data Availability

The data that support the findings of this study are available on request from the corresponding author.

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
