# Peer review of "Effects of Exercise on Positive Symptoms, Negative Symptoms, and Depression in Patients with Schizophrenia: A Systematic Review and Meta-Analysis"

_ijerph, 2023, doi:10.3390/ijerph20043719_

Round 1

Reviewer 1 Report

The results of the analyzes presented in the article may bring specialists closer to solving the important problem of increasing the effectiveness of therapy for people with schizophrenia.

The article presents a meta-analysis of data presented in selected papers - according to specific criteria.

In search of articles on the importance of physical exercise for the intensification of positive and negative symptoms and depression in patients diagnosed with schizophrenia, the PubMed, Embase, Cumulative Index of Nursing and Allied Health Literature (CINAHL), MEDLINE, Cochrane Central Register of Controlled Trials, PsycINFO databases were used, Web of Science, and Google Scholar. Database searches were limited to the last ten years.

As a result, 15 studies conducted in 10 countries were included in the meta-analysis. (On page 4, the authors show 14 studies - 149-152. The difference is due to the omission of one US study. The rest of the article mentions 17 studies. It would be good to check the number of studies included in the meta-analysis.) Noteworthy is the proportion of studies from Asian countries (8) and from other parts of the world that were included in the meta-analysis. I am not talking about "geography" but about the cultural context: the importance of physical activity, and various types of exercise common in different cultures for maintaining psychophysical conditions. It can be expected that this is important for the approach to the treatment of both medical teams and patients and their families: traditional for the "Western" culture (medical) or holistic - and beliefs about the greater or lesser effectiveness of exercise/physical therapies.

This criterion was not included in the analysis presented in the article, and the results of the analysis in this respect could be interesting. Perhaps researchers will be inspired to conduct such an analysis - in this or another paper.

The small number of studies included in the meta-analysis is due to their limitations or shortcomings recognized by the authors. This is an important point made by the authors.

The authors presented the course of selection, and the results of the analysis and discussed them referring to other researchers. Some of the studies cited in the article (in the discussion) brought similar results to those obtained by the authors - in terms of the effectiveness of various types of physical exercise for changes in the severity of positive and negative symptoms and depression in people diagnosed with schizophrenia (and schizoaffective disorder), and others - different. This justifies the need for further exploration of therapy for people with symptoms of schizophrenia.

I am wondering about the legitimacy of selecting studies based on other criteria, including a more precise determination of the age range of patients, period, pharmacotherapy, and psychophysical condition of patients at the time of conducting exercise therapy (and tests/measurements). Would the results of the meta-analysis be similar in terms of the usefulness of this type of therapy for changing the severity of positive and negative symptoms in people with schizophrenia?

The article may be of interest to specialists, if only because of the results regarding the effectiveness of therapy other than pharmacotherapy.

Author Response

We would like to express our sincere thanks to you and the reviewers for your thorough review of our manuscript and for the opportunity to submit a revised and improved version. We believe that by addressing the concerns, we have considerably improved our manuscript.

Please find below our point-by-point responses to the reviewers’ comments. The changes made in the revised manuscript are marked in red bold text. We hope that this revised manuscript (Manuscript ID: IJERPH-2188680) is suitable for publication in the International Journal of Environmental Research and Public Health. We will certainly be willing to make additional changes should they be required.

I attached the file (Response to Reviewers' Comments).

Reviewer 2 Report

The manuscript reports findings of a meta-analytic review on the efficacy of exercise on improvement in positive, negative, and depressive symptoms in individuals with schizophrenia. The quantitative synthesis of findings from randomized controlled trials has offered potentially compelling evidence for the efficacy of exercise in the reduction of these symptoms. However, I would propose an expansion on the scientific value of the investigation of the effect of exercise on schizophrenia symptoms, which could be further beefed up in the Introduction and the Discussion. The authors may consider supplementing details of the exercise interventions, for example, types and contents, as well as the potential underlying mechanisms leading to the improvement in positive, negative and depressive symptoms. The authors decided to consider separate symptom dimensions, which should be justified by the theoretical formulation that exercise may have differential impacts and mechanisms of changes on these dimensions. This would set the stage for the quantitative syntheses of the findings of the included studies, paving way for the elaboration on the results of the meta-analyses and discussion (e.g. treatment implications) in the Discussion section, which is a bit descriptive in this point. 

My further comments are as follows:

Abstract:

- In line 17, “random effect model” should be revised as “random-effects model”.

Introduction:

- The authors argued that existing reviews had several limitations, which are addressed in the current review (lines 63-75, p.2). Some of the statements need to be strengthened and further clarified. In particular, for the statement “did not have sufficient reliability to reveal the effectiveness” (line 69, p. 2), do the authors suggest that the number of the included studies in previous reviews was small, or the sample size of the individual included studies was not adequately powered? I would recommend highlighting the inclusion of only RCT studies, as a more stringent inclusion criterion in this review, as a new contribution to this work. Some arguments in the first paragraph of the Discussion (lines 306-315, p. 12) would be highly relevant and should be presented earlier in the Introduction. 

 Methods:

- I am curious about the inclusion of psychiatric diagnosis in this review, as samples with patients with schizoaffective disorder were also included, in addition to schizophrenia (while the title suggested that only schizophrenia was included in this review). Were samples with participants with other schizophrenia spectrum disorders, such as delusional disorder or psychosis NOS, excluded? Was the decision of this criterion pre-determined or made ad-hoc based on the included studies after the screening?

- The cutoff of various levels of the size of the SMD and also statistical heterogeneity should be mentioned here, as the descriptors like “moderate heterogeneity” and “medium/ small effect size” were used in the Results and Discussion.

- For statistical heterogeneity, any rationale on why 0.1 (instead of the standard practice of 0.05) was chosen as a cut-off for the p-value to determine statistical significance (ref line 136, p. 3)?

 Results:

-          Referring to lines 175 – 178 (p. 4) where the summary measures of the symptoms of the included studies were reported, I suggest adding the symptom measures of each study in Table 1 so readers could know the exact measures used in these studies. Also, for depressive symptoms, is only item G6 from the PANSS (instead of the whole scale) was extracted and included in the effect size calculation?

- In sections 3.4.1, 3.4.2 and 3.4.3, when discussing the largest effect size among the included studies, please also add the citation of the study for readers’ reference (although the forest plots may have offered a clue).

-          The manuscript did not include an analysis of publication bias for the meta-analysis of depressive symptoms. Also, the sensitivity analysis for the meta-analyses of positive and depressive symptoms seemed missing. These should be conducted and reported for the completeness of the analyses.

 Discussion:

- I am not convinced by the statement “This result revealed that the effect on positive symptoms was smaller than that on negative symptoms” (lines 340-341, p. 13). After adjustment of publication bias, the pooled effect size for negative symptoms diminished (0.28 (95% CI: -0.52 to -0.04), which its CI overlaps with that of positive symptoms (-0.24 (95% CI: -0.43 to -0.04). This may suggest that their effects do not differ.

 Other minor points:

- Some of the citations did not match the cited work in the reference list. Please double-check.

- The long form of ACSM should be spelt in full when it first appears in the manuscript.

Author Response

(The authors gave the same response as above.)

Round 2

Reviewer 2 Report

Dear Authors,

Thanks for the effort to address my comments and revise the manuscript accordingly. The manuscript is now much improved, in terms of clarity and value of the research question. I just have three comments (the second and third were raised in my previous report but might have been missed) for your consideration:

 Introduction:

-        The argument for the impact of exercise on improvement is much more explicit and clearer now. I would further suggest adding a statement or two to talk about the potential differential impact of exercise on positive and negative symptoms and their underlying mechanisms (which might be distinct), supporting separate analyses on their own. Some of the arguments in the Discussion, e.g. in lines 409-414 (p. 13), could be moved here to let readers understand the need to consider positive and negative symptoms separately earlier.

-        Referring to lines 196-198 (p. 6) where PANSS was mentioned to be a measure of depressive symptoms, please confirm if only item G6 from the PANSS (instead of the whole scale) was extracted and included in the effect size calculation for the meta-analysis of depressive symptoms?

-        The manuscript did not include an analysis of publication bias for the meta-analysis of depressive symptoms. Also, the sensitivity analysis for the meta-analyses of positive and depressive symptoms seemed missing. These should be conducted and reported for the completeness of the analyses.

Author Response

(The authors gave the same response as above.)
